# Characteristics of Psychosocial Factors in Liver Transplantation Candidates with Alcoholic Liver Disease before Transplantation: A Retrospective Study in a Single Center in Taiwan

**DOI:** 10.3390/ijerph17228696

**Published:** 2020-11-23

**Authors:** Yu-Ming Chen, Tien-Wei Yu, Chih-Chi Wang, Kuang-Tzu Huang, Li-Wen Hsu, Chih-Che Lin, Yueh-Wei Liu, Wei-Feng Li, Chao-Long Chen, Chien-Chih Chen

**Affiliations:** 1Department of Psychiatry, Kaohsiung Chang Gung Memorial Hospital and Chang Gung University College of Medicine, Kaohsiung 833, Taiwan; yuming0320@gmail.com (Y.-M.C.); mp9596@cgmh.org.tw (T.-W.Y.); 2Liver Transplantation Center and Department of Surgery, Kaohsiung Chang Gung Memorial Hospital and Chang Gung University College of Medicine, Kaohsiung 833, Taiwan; ufel4996@ms26.hinet.net (C.-C.W.); hsuliwen1230@gmail.com (L.-W.H.); chihchelin@cgmh.org.tw (C.-C.L.); anthony0612@cgmh.org.tw (Y.-W.L.); webphone@cgmh.org.tw (W.-F.L.); 3Institute for Translational Research in Biomedicine, Kaohsiung Chang Gung Memorial Hospital and Chang Gung University College of Medicine, Kaohsiung 833, Taiwan; huangkt@cgmh.org.tw

**Keywords:** alcoholic liver disease, liver transplantation, depression, psychosocial evaluation, family function

## Abstract

Liver transplantation (LT) is an essential treatment for end-stage alcoholic liver disease (ALD). The patients’ psychosocial condition plays a vital role in post-transplantation prognosis. A survey of the candidates’ psychosocial wellbeing is necessary before LT. This study aims to investigate the psychosocial characteristics, including the depression degree, family function, alcohol use duration, and alcohol abstinence period, of LT candidates with ALD. In addition, 451 candidates for LT due to ALD were enrolled. They received psychosocial evaluations, including depression scale (Hamilton depression rating scale) and family functioning assessment (adaptability, partnership, growth, affection, resolve (APGAR) index). The test scores were analyzed according to age, alcohol use duration, and alcohol abstinence period. The Hamilton depression rating scale (HAM-D) score and the family APGAR index score differentiated significantly according to the age, alcohol use duration, and abstinence period of the LT candidates. The patients with shorter alcohol use duration tended to have more severe depressive symptoms and poorer family support. The younger patients showed a significantly shorter abstinence period, more severe depression, and poorer family functioning than older patients. The younger ALD patients and patients with shorter alcohol use duration showed an increased severity of depression before transplantation. They need more mental health care over time.

## 1. Introduction

Liver transplantation (LT) is considered a treatment option for patients with liver dysfunction in end-stage liver disease. Indications for LT are various and include end-stage liver cirrhosis, hepatocellular carcinoma, fulminant liver disease, hepatitis C virus, and alcohol liver disease (ALD). LT has been performed increasingly in Europe and Asia [1,2]. One-quarter of the liver transplantations in the USA each year are due to alcohol liver disease [3]. Due to the limited availability of liver donation, the prioritization of LT waiting lists should be evaluated fairly based on who is most likely to benefit from LT [4].

In the early 1980s, many studies indicated that ALD patients who underwent LT had a poor prognosis and worse disease course than non-ALD patients [5,6,7]. At the same time, stricter criteria have been developed for waiting lists. A previous study about LT and alcohol use pointed out higher relapse rates in patients who have less than a 6 month period of abstinence [8]. A relationship between the period of alcohol abstinence and the risk of relapse of alcohol use has been clearly found [9,10], and the “6 month rule” is broadly applied as an important precondition for ALD candidates for LT. However, various studies showed good survival rates of ALD patients and grafts after LT, and the 5 year survival rates are greater than 75% [11]. Recently, the post-transplantation prognosis and clinical course of ALD patients who had no alcohol relapse after transplantation were shown to not be significantly different from those of transplanted patients with other etiologies of liver disease [8,12,13]. Nevertheless, approximately 20%–25% of ALD patients who receive LT relapse drink excessively [10], which may lead to a higher risk of liver impairment and mortality afterward [14,15].

As a consequence, several studies of alcohol use and the relapse after transplantation have been performed. Potts reported an increased risk of psychosocial impairment in ALD patients compared with non-ALD patients [16]. Multiple psychosocial factors are linked with alcoholism and may exacerbate further prognosis. Common risk factors for relapse are poor social or family support, short duration of pre-transplantation abstinence, and anxiety or depressive disorder [17]. A comprehensive survey of related risk factors and the reinforcement of social or family support may reduce the relapse rate of these patients.

Depressive symptoms are common among individuals with alcoholism. A recent study indicated a causal relationship between depression and alcohol use disorder, and the risk of depression may increase with the degree of alcohol consumption [18]. Lai et al. found an increased risk of the comorbidity of anxiety or mood disorders with ALD in a meta-analysis [19]. Hassan et al. suggested a bidirectional relationship in which, on the one hand, patients may use alcohol as “self-medication” to alleviate depression, but on the other hand, long-term alcohol use may contribute to more severe depressive symptoms, which could be caused by impaired social relationships and/or direct alcohol damage to the brain [20]. Uncontrolled alcohol consumption can also cause other problems, such as familial relationship distress, domestic violence, trauma, and motor vehicle accidents. In addition to depression, family and social support also play a role in ALD patients. Risky environments, such as poor social support, stressful life events, family violence, and childhood maltreatment, may increase the risk of excessive alcohol drinking, conduct problems, and depressive symptoms when the genetic vulnerability is also present [21]. The research found that social and family support are associated with susceptibility to the development of alcoholism in both adolescents and adults, alcohol use initiation, the onset of problematic behavior related to alcohol, and the maintenance of alcohol intake [22,23]. Averna et al. reported that peer social and family support are related to drug and alcohol use in adolescents [24].

Several studies have been performed regarding the psychosocial factors of liver donors [25,26,27,28], but the psychosocial functioning of LT recipients with ALD has drawn little attention. Pegum et al. found a decreased severity of depression and anxiety and improved quality of life among ALD patients after they received LT [26]. In a survey study, LT recipients with ALD demonstrated an obvious burden of substance use, mental health symptoms, and disability [29]. As there is less focus on the psychosocial functioning of patients with ALD, this retrospective study aims to investigate the relationship between psychosocial factors (depression status and family support) of LT recipient candidates, and alcohol use duration and alcohol remission status. The other objective of this study is to better understand the psychosocial impairment of LT recipient candidates with ALD to promote comprehensive mental health care.

## 2. Materials and Methods

### 2.1. Study Design

We collected data from LT candidates with ALD at the Liver Transplant Center of Kaohsiung CGMH before liver transplant. The Institutional Research Ethics Committee of CGMH has approved the study (Approval No. CGMH 201900889B0). We performed a retrospective review of the preoperative chart data from LT candidates. The data were gathered during the routine psychiatric interdisciplinary care, and the confidentiality of the data was protected. There were 451 candidates who met the inclusion criteria, which included enough language ability and residency in Taiwan. The data collection period was between January 2012 and May 2019. Demographic data including age, education degree, occupation, and marital status were collected. Alcohol consumption status, including family history of alcoholism, duration of the consumption, abstinence duration, and consumption amount, are recorded. The patients’ past psychiatric history, such as mood disorder, anxiety disorder, formal thought problems, and cognition impairment, is also documented. The depressive symptoms of the LT recipient candidates were evaluated through the Chinese version of the Hamilton depression rating scale (HAM-D). Family functioning was assessed with the aid of the Chinese version of the family APGAR index (adaptability, partnership, growth, affection, resolve). The questionnaires were administered at the hospital by the psychiatrist.

### 2.2. Questionnaires

#### 2.2.1. HAM-D

The Hamilton depression rating scale (HAM-D) is a multiple-item questionnaire that is broadly used to rate the severity of depression. The HAM-D has been found to have good reliability and fair validity [30]. The severity of depression in patients is evaluated by a physician on the basis of 17 items in areas including mood, somatic symptoms, guilty feeling, weight loss, suicidal ideation, anxiety, insomnia, psychomotor agitation, and retardation. Each item is assessed on a three- or five-point scale. The severity of depression is classified as follows: Scores < 7 = no depression, scores ≥ 17 = mild depression, and scores ≥ 24 = moderate to severe depression.

#### 2.2.2. Family APGAR Index

The family APGAR index has been widely utilized to investigate the relationship of family functioning by measuring an individual’s perception of their family functionality [31,32]. The family APGAR index is a self-report survey that contains subdomains of adaptation, partnership, growth, affection, and resolve. This index is determined with the use of a five-item questionnaire. The total score is calculated by adding the values of the five items. Family functioning is determined according to the total score as follows: Score 7–10 = good, score 4–6: Moderate dysfunction, score 0–3: Severe dysfunction.

### 2.3. Statistical Analysis

The collected data were evaluated using the Statistical Package for Social Sciences software version 17 (SPSS, Inc., Chicago, IL, USA). The data are presented as the means and the standard deviations for the analysis of descriptive statistics. The differences in the mean in the form of continuous variables were examined using Spearman’s correlation with significance tests. The related variables were analyzed as influencing factors of psychosocial stress in LT candidates, including demographic data (gender, education level, occupational status, marital status, family history of alcoholism) and alcohol use (alcohol use duration, alcohol abstinence period). For the further detailed survey, variables that showed significant differences according to the HAM-D score and the family APGAR index score in the univariate analyses were recruited in the multivariate analyses. We performed multiple linear regression to assess independent factors of LT candidates’ psychosocial and family functionality.

## 3. Results

### 3.1. LT Recipient Candidates’ Demographic Characteristics

The demographic characteristics of the LT recipient candidates with ALD are listed in Table 1. There were 451 participants enrolled in this study (6.9% female and 93.1% male), and the mean age was 50.6 years old. In the study samples, 65 (14.4%) had an education level of primary school, 333 (74%) finished senior or junior school, and 52 (11.6%) graduated from college or above. Regarding the marital status of recipients, 35 (7.8%) were single, 364 (80.9%) were married, and 51 (11.3) were divorced or widowed. It was determined that 49 (10.9%) patients were unemployed, 365 (80.9%) were employed or were housewives, and 37 (8.2%) were retired.

### 3.2. Correlations Between Alcohol Use and Psychosocial Functions

We examined the correlations between age, alcohol use duration, alcohol abstinence period, depression, and family function (Table 2). The younger patients showed significantly shorter abstinence periods (r: 0.188, *p* < 0.001) and an increased severity of depression (r: −0.155, *p* = 0.001) compared with the older patients. In addition, the patients with shorter alcohol use duration tended to have more severe depressive symptoms (r: −0.137, *p* = 0.004) and poorer family support (r: 0.123, *p* = 0.009).

Besides, better family functioning was associated with longer abstinence periods (r: 0.102, *p* = 0.034) and longer alcohol use durations (r: 0.123, *p* = 0.009). A strong negative correlation was found between the HAM-D score and the family APGAR index score (r: −0.249, *p* < 0.001). The summarized correlation analysis is presented in Table 2.

### 3.3. Univariable and Multivariable Analysis of Factors Influencing Depression and Family Functioning

We examined the patients’ characteristics that influenced the HAM-D scores, and the family APGAR scores. Table 3 presents the results of the simple linear regression and multiple linear regression analysis. Regarding the depression degrees measured with the HAM-D score, gender, age, education level, occupational status, alcohol use duration, and alcohol abstinence duration played a role in depression severity significantly. On the other hand, family function measured with APGAR score was affected by age, occupational status, marital status, family history of alcoholism, alcohol use duration, and abstinence duration significantly. Next, we conducted the multiple linear regression with the stepwise method for the HAM-D and APGAR scores. The multivariate approach for developing a prediction model was also reported in a recent research article [33]. The adjusted R square in multiple linear regression for HAM-D score was 0.054 and for APGAR score was 0.256.

### 3.4. Occupational and Marital Statuses in Relation to Alcohol Use Duration

When the alcohol use duration of the LT recipient candidates with ALD was examined, the married patients had the longest duration of alcohol use (*p* = 0.035), followed by divorced/widowed patients, and the single patients had the shortest duration of alcohol use (*p* = 0.002) (Table 4).

We also discovered that, regarding patients’ occupational status, retired patients had the longest duration of alcohol use, followed by unemployed patients (*p* = 0.001), and employed patients/housewives (*p* < 0.001) had the shortest alcohol use duration (Table 4).

## 4. Discussion

ALD is a common indication for LT in Europe and the United States and is responsible for approximately 15%–20% of the LTs performed [34,35,36]. Liver transplantation for ALD patients is a controversial issue because of the limited number of donated organs, and relapse would be harmful to the donated liver. However, current research on addictive behavior indicated that alcohol abuse is a chronic disease of the brain and has a genetic component [37,38]. In light of ethical issues, these patients should be treated equally without discrimination [39,40,41,42]. Given the multifaceted nature of ALD and transplantation processes, the pre-transplantation evaluation will be essential to determine the appropriateness of transplantation and to set a feasible plan before and after LT [43,44,45]. The work of such a multidisciplinary team in transplant centers has shown declined rates of alcohol relapse and mortality after transplantation [34,46]. There are several psychosocial factors that are supposed to be predictors of relapse, such as the duration of abstinence, social support, family history of alcoholism, a history of severe mental disorder, and the duration of alcoholism [17].

In this retrospective study, we found that patients with shorter alcohol use durations showed significantly more severe depressive symptoms than those with longer alcohol use durations. The higher depression grade in candidates with shorter alcohol use duration might be due to probable recent stressful life events and inadequate stress-coping strategies, such as binge drinking. Otherwise, the younger patients showed a significantly shorter abstinence period and an increased severity of depression compared with the older patients. One possible explanation is that young people tend to consume alcohol in the pattern of binge drinking [47], which may contribute to an exacerbation of depression. Powers et al. demonstrated that binge drinking behavior in young women appears to enhance the risk of depressive symptoms [48]. Furthermore, binge drinkers presented higher suicidal ideation, and suicidal plans and attempts than non-binge drinkers [49]. Therefore, younger candidates and candidates with shorter alcohol use durations need more intensive treatment for depression. The less depressive symptoms in candidates with a long-term drinking history might be explained by the fact that they drink only habitually from the beginning rather than drinking as a stress reliever. Nevertheless, this pattern of drinking leads to a transition from casual drinking to addiction. A study of ALD patients undergoing LT demonstrated that approximately 50% of patients experienced depressive symptoms [29]. The identification of depressive symptoms without delay and prompt treatment are important for patients.

As anticipated, a better family APGAR index score was associated with a lower HAM-D score in our study, illustrating that better family support was related with less depression in patients with ALD. In the post-transplant phase, social support was thought to be a vital factor in preventing alcohol abuse [50,51]. Good support from family and a reliable partner could be a protective factor to reduce rates of relapse; in contrast, poor social stability and social activities in an alcohol-related event may increase the risk of relapse [52]. However, better family functioning was related to longer alcohol use duration in our study. One possible reason is that, compared to binge drinkers, the family members of non-binge drinkers may get used to the habitual drinking of the patients over a long period of time, and they may develop tolerance to the patient’s behavior and could provide sustained family support. The exact causal relationship between family support and drinker types deserves more evaluation.

The multiple regression analysis in this study indicated that education level had a significant association with both depression severity and family functioning in ALD patients. The patients with lower educational levels suffered from more depressive symptoms and less family support than patients with college/above degrees, and the results demonstrated a significant difference in the senior/junior school group and a mild difference without statistical significance in the primary school group. The role of education level in alcohol abuse is controversial. Torikka et al. described that adolescents with lower education levels and who were unemployed presented an increased frequency of drinking and drunkenness [53]. By contrast, another study showed that adolescents with high education levels predicted more frequent alcohol use and binge drinking [54]. The interaction among family functioning, education level, and depression is complex and needs further investigation.

Our data demonstrated that married patients with ALD had longer alcohol use duration in comparison to divorced/widowed patients and single patients. Marital status seems to influence drinking behavior. Correine et al. surveyed the influence of marital status on alcohol abuse among older adults, and the data showed that being married/remarried increases women’s drinking but reduces men’s alcohol consumption in comparison to never being married [55]. Otherwise, our data showed that retired patients had a longer alcohol use duration, followed by unemployed patients and employed patients/housewives. Nicholson et al. indicated that retired people used alcohol in various patterns and that drinking could be linked to relaxation and active social engagement, while harmful drinking is associated with social isolation [56]. Regarding employment status, an epidemiological study has shown that income inequality is correlated with serious mental illness [57]. Similarly, the association of income inequality with a greater incidence of depression in women was found in the United States [58]. Additionally, Park et al. reported that occupational status, such as occupational type, working hours per week, working schedule, and working status, is correlated with depression among older Koreans [59]. In the working population, a more supportive work environment, less stress in work, and higher prestige are protective factors against depression [60]. In general, occupational status affects the patients’ life in various domains including mental wellness, and thus, it is important to evaluate the patients’ employment status in pre-operative surveys. Different occupational and marital statuses may affect alcohol use patterns. The drinking pattern deserves research to develop public health policies that support preventing harmful drinking.

This study has limitations. First, our study did not determine the time sequence of the diagnosis of depression and alcohol consumption. A patient who had depressive disorder before they started to consume alcohol may present more severe depressive symptoms [18], which affects our assessment of the severity of depression in patients. Second, we examined some common psychosocial factors of ALD patients, such as education level, occupational status, marital status, depressive symptoms, and family functions. However, other psychosocial dimensions, such as anxiety, stress, somatization, social network/support, and life events, also play a role in the wellbeing of patients. Finally, our survey was designed as a cross-sectional study. We did not perform the post-transplantation follow-up of patients.

This study emphasized the importance of the pre-transplantation evaluation of psychosocial factors in ALD patients. According to the results, younger ALD patients showed shorter alcohol use durations and shorter abstinence periods, were exposed to an increased risk of depressive symptoms, and had impaired family functioning. These patients need adequate and timely intervention for depression and family dysfunction. On the other hand, older ALD patients had longer alcohol use durations and longer abstinence periods, showed less depression, and had better family support. We should pay attention to the consequences of long-term alcohol toxicity and severe physical problems, such as liver cirrhosis or even liver malignancy. It seems that drinker type of habitual drinking or binge drinking might have different influences on the psychosocial functioning of ALD patients and deserves further exploration.

## 5. Conclusions

The younger patients with ALD and patients with shorter alcohol use duration suffered from more severe depression and poorer family support before LT. As comorbid psychiatric disease was associated with higher relapse rates after LT, which may lead to graft dysfunction, prompt identification and treatment for depression are necessary for the patients. Although the selection of patients for transplantation was not influenced by age or comorbid depression, we should look after these patients following LT. Education of patients and their family members about regular outpatient follow-up, adequate compliance of immunosuppressants, and enhancement of a healthy lifestyle are essential. Communities or social organizations (such as alcoholics anonymous), and psychological intervention, including motivational enhancement therapy, cognitive behavioral therapy, and twelve-step facilitation, are very helpful for supporting the patients on the route to recovery.

## Figures and Tables

**Table 1 ijerph-17-08696-t001:** The demographic characteristics of the liver transplantation (LT) recipient candidates.

Variable	Total (n = 451) (n [%])
Age (y/o) (mean ± SD (range))	50.6 ± 7.8 (29–69)
Female	51.5 ± 7.4 (36–66)
Male	50.5 ± 7.9 (29–69)
Sex	
Female	31 (6.9)
Male	420 (93.1)
Education level ^1^	
Primary school	65 (14.4)
Senior/junior school	333 (74)
College	52 (11.6)
Occupational status	
Unemployed	49 (10.9)
Employed/housewife	365 (80.9)
Retired	37 (8.2)
Marital status ^1^	
Single	35 (7.8)
Married	364 (80.9)
Divorced/widowed	51 (11.3)
Family history of alcoholism ^1^	
None	240 (53.3)
Immediate family	122 (27.1)
Siblings	88 (19.6)

Note: ^1^ Missing data: 1.

**Table 2 ijerph-17-08696-t002:** Spearman’s correlations for psychosocial functioning and alcohol use (n = 451).

Variable	Age (Years Old)	Alcohol Use Duration (years)	Abstinence Duration (months)	HAM-D Scores	APGAR Scores
Age (years old)	1				
Alcohol use duration (years)	0.809 **	1			
Abstinence duration (months)	0.188 **	0.185 **	1		
HAM-D scores	−0.155 **	−0.137 *	−0.076	1	
APGAR scores	0.204 **	0.123 *	0.102 *	−0.249 **	1

Note: * *p* < 0.05; ** *p* < 0.001.

**Table 3 ijerph-17-08696-t003:** The evaluation of Hamilton depression rating scale (HAM-D) scores; adaptability, partnership, growth, affection, resolve (APGAR) scores; and LT patients’ characteristics: Univariable and multivariable analysis.

Variable	HAM-D Scores				APGAR Scores			
	Simple Linear Regression		Multiple Linear Regression		Simple Linear Regression		Multiple Linear Regression	
	B (95% C.I.)	*p*-Value	B (95% C.I.)	*p*-Value	B (95% C.I.)	*p*-Value	B (95% C.I.)	*p*-Value
Sex ^1^	1.46 (0.47, 2.45)	0.004 *	−1.53 (−2.56, −0.49)	0.004 *	−0.67 (−1.40, 0.06)	0.075		
Age (years old)	−0.04 (−0.08, −0.01)	0.003 *	−0.03 (−0.07, −0.00)	0.035 *	0.05 (0.03, 0.07)	0.000 **		
Education level ^2^								
Primary school	0.43 (−0.54, 1.42)	0.383			0.05 (−0.67, 0.79)	0.877		
Senior/junior school	1.09 (0.29, 1.88)	0.007 *	0.79 (0.19, 1.39)	0.010 *	−0.57 (−1.15, 0.01)	0.058	−0.57 (−0.94, −0.19)	0.003 *
Occupational status ^3^								
Unemployed	1.26 (0.10, 2.42)	0.033 *			−1.07 (−1.93, −0.21)	0.015 *		
Employed/ housewife	0.85 (−0.06, 1.77)	0.068			−0.35 (−1.03, 0.33)	0.311		
Marital status ^4^								
Single	−0.02 (−1.18, 1.13)	0.961			0.77 (0.00, 1.54)	0.050 *	0.86 (0.10, 1.63)	0.027 *
Married	−0.43 (−1.22, 0.35)	0.284			2.70 (2.18, 3.22)	0.000 **	2.76 (2.24, 3.28)	0.000 **
Family history of alcoholism ^5^								
None	−0.00 (−0.67, 0.66)	0.985			0.59 (0.11, 1.08)	0.016 *	0.47 (0.14, 0.80)	0.005 *
Immediate family	−0.18 (−0.93, 0.56)	0.626			0.19 (−0.35, 0.74)	0.481		
Alcohol use duration (years)	−0.036 (−0.06, 0.00)	0.013 *			0.026 (0.006, 0.047)	0.013 *		
Abstinence duration (months)	−0.006 (−0.01, 0.00)	0.034 *			0.005 (0.000, 0.009)	0.032 *		

Note: ^1^ Compared to male, ^2^ Compared to college/above, ^3^ Compared to retired, ^4^ Compared to divorced/widowed, ^5^ Compared to sibling, * *p* < 0.05, ** *p* < 0.001. Adjusted R square in multiple linear regression for HAM-D score: 0.054; for APGAR score: 0.256.

**Table 4 ijerph-17-08696-t004:** Univariable analysis of association between alcohol use duration and marital status/occupational status.

Variable	Alcohol Use Duration (Years)	
	B (95% C.I.)	*p*-Value
Marital status		
Single vs. divorced/widowed	−5.987 (−9.751, −2.222)	0.002 *
Married vs. divorced/widowed	2.763 (0.198, 5.328)	0.035 *
Occupational status		
Unemployed vs. retired	−6.256 (−10.018, −2.495)	0.001 **
Employed/housewife vs. retired	−7.822 (−10.802, −4.842)	<0.001 **

Note: * *p* < 0.05, ** *p* < 0.001.

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
