# Peer review of "Characteristics of Psychosocial Factors in Liver Transplantation Candidates with Alcoholic Liver Disease before Transplantation: A Retrospective Study in a Single Center in Taiwan"

_ijerph, 2020, doi:10.3390/ijerph17228696_

Round 1
Reviewer 1 Report
This manuscript was important in terms of deciding the indication for liver transplantation. We really worry whether we should conduct liver transplantation in case of considering recipient’s background especially in living donor liver transplantation. I have some comments.
- Why did younger patients with shorter addiction of alcohol have severe depression? These patients with severe liver dysfunction lost their hope for the future and their despair caused depression?
- Was there positive relationship between the severity of liver dysfunction (MELD score or Child –score) and depression score (HAM-D score and APGAR score)?
- Why didn’t they exclude the patients with longer period of alcohol addiction from the criteria of liver transplantation because these patients will tend to relapse an addiction after liver transplantation?
Reviewer 2 Report
The authors report interesting data regarding a cohort of candidate patients for liver transplantation. However, there are several concerns in terms of statistical analysis. Tables will be pointed out, however the changes should be addressed also in the results and discussions sections:
- Table 2: The authors employ the Spearman's correlation test to determine correlations between continuous and dichotomous variables. In this context the used test is not correct (Spearman and Pearson can be used for a pair of continuous variables).
- Table 3: the authors report a "multiple" linear regression. To conduct a multiple linear regression it is not possible to feed the data to the program. Data should be analyzed univariately and then multivariately by choosing the best multivariate model according to the model calibration metrics (AIC; BIC; etc..). In this regards I suggest to check the statistical analysis and cite this recent paper on another MDPI journal: https://doi.org/10.3390/diagnostics10090619
- Table 4: are odds ratio studied univariately or multivariately?
Reviewer 3 Report
This is a really interesting paper, investigating an important population and potentially with important implications for practice in relation to allocation of people for liver transplantation and support for people through the process, with good indications about special needs of young people and people who have been drinking over shorter periods of time.
It is, however, severely let down at the moment by the Conclusions section, which needs to draw on the richness and complexity of the investigation and discussion. That should be easy to do. What conclusions do you draw about implications for clinical practice, allocations for transplant, social support, engagement of families and communities in supporting abstinence, aftercare?
One other issue that would have been useful, if the data has been gathered, would have been to know more about types of employment and any links betwen those and psychosocial status, given that there is a wide range of literature - eg Richard Wilkinson et al - suggesting that employment status has a major psychosocial impact
Round 2
Reviewer 2 Report
The authors have edited the manuscript according to the reviewer's request. The manuscript can be accepted in its current form.
